# Incidence and Management of Hypertriglyceridemia-Associated Acute Pancreatitis: A Prospective Case Series in a Single Australian Tertiary Centre

**DOI:** 10.3390/jcm9123954

**Published:** 2020-12-06

**Authors:** Hong Lin Evelyn Tan, Georgina McDonald, Alexander Payne, William Yu, Zahrul Ismadi, Huy Tran, Jon Gani, Katie Wynne

**Affiliations:** 1Department of Diabetes & Endocrinology, John Hunter Hospital, New Lambton Heights, NSW 2305, Australia; honglinevelyn.tan@health.nsw.gov.au; 2School of Medicine & Public Health, University of Newcastle, New Lambton Heights, NSW 2305, Australia; georgina.mcdonald@health.nsw.gov.au (G.M.); huy.tran@health.nsw.gov.au (H.T.); jon.gani@health.nsw.gov.au (J.G.); 3Department of Medicine, John Hunter Hospital, New Lambton Heights, NSW 2305, Australia; alexander.payne@health.nsw.gov.au (A.P.); william.yu@health.nsw.gov.au (W.Y.); 4Department of Clinical Chemistry, John Hunter Hospital, New Lambton Heights, NSW 2305, Australia; zahrul.ismadi@health.nsw.gov.au; 5Department of Surgery, John Hunter Hospital, New Lambton Heights, NSW 2305, Australia

**Keywords:** acute pancreatitis, hypertriglyceridemia, intravenous insulin therapy

## Abstract

Background: Hypertriglyceridemia-associated acute pancreatitis (HTGAP) has been linked with increased severity and morbidity. In this study, triglyceride levels were measured in all patients admitted with acute pancreatitis (AP) to establish the incidence of HTGAP in an Australian center. Methods: A prospective cohort with AP was collated over an 18-month period in a single tertiary referral hospital. HTGAP was defined as AP with triglycerides ≥ 11.2 mmol/L (1000 mg/dL). Incidence, clinical co-morbidities, severity and management strategies were recorded. Results: Of the 292 episodes of AP, 248 (85%) had triglycerides measured and were included. HTGAP was diagnosed in 10 of 248 (4%) AP cases. Type 2 diabetes, obesity, alcohol misuse and gallstones were common cofactors. The HTGAP group demonstrated severe hypertriglyceridemia compared to the non-HTGAP group (median 51 mmol/L vs. 1.3 mmol/L). Intensive care unit (ICU) admissions were significantly increased (odds ratio (OR) 16; 95% CI 4–62) in the HTGAP group (5/10 vs. 14/238 admissions, *p* < 0.001) and constituted 26% (5/19) of total ICU admissions for AP. Four patients received intravenous insulin with fasting and had a rapid reduction in triglyceride levels by 65–77% within 24 h; one patient had mild hypoglycemia secondary to therapy. Conclusion: HTGAP occurred in 4% of AP cases and was associated with higher risk of ICU admission. Intravenous insulin and fasting appear safe and efficacious for acutely lowering triglyceride levels in HTGAP.

## 1. Introduction

Hypertriglyceridemia-associated acute pancreatitis (HTGAP) is thought to account for 0.7–20% of acute pancreatitis (AP) cases and is associated with increased severity and morbidity [1,2,3,4,5,6,7,8,9,10]. The triglyceride threshold thought to be causal of AP is considered as ≥11.2 mmol/L (1000 mg/dL), though this number is somewhat arbitrary [11,12]. The absolute risk of AP is predicted to be 5% in severe hypertriglyceridemia (HTG) ≥ 11.2 mmol/L (1000 mg/dL) and up to 10–20% in very severe HTG ≥ 22.4 mmol/L (2000 mg/dL) [2,8].

The potential mechanism of pancreatitis in hypertriglyceridemia remains unclear and largely based on animal studies [13,14]. Havel’s theory is well-described, in which hydrolysis of excess triglycerides by pancreatic lipase increases free fatty acid (FFA) concentrations, exceeding the plasma albumin-binding capacity. Excess FFAs self-aggregate to form micellar structures that act as detergents within an acidic microenvironment; these micelles compromise the integrity of pancreatic acinar cells and pancreatic capillaries, causing inflammation and ischemia [15]. Hyperviscosity due to high concentrations of larger chylomicrons is also thought to impede blood flow to the pancreas, leading to further ischemia. This causal role for FFA was demonstrated in canines, where infusion of FFAs into the canine pancreas directly led to oedema and hemorrhage [14]. An underlying genetic predisposition has also been proposed; a Spanish study has shown an increased risk of HTGAP in people with apolipoprotein E ε4 allele; a Chinese study has shown an increased risk in people with the cystic fibrosis transmembrane conductance regulator mutation/variant/haplotype with a tumor necrosis factor promoter polymorphism [16,17].

If, indeed, hypertriglyceridemia is pathological in humans, whether and how to intervene remains unclear, with ongoing debate around the use of intravenous insulin infusion and lipid apheresis [18].

This study prospectively examines the incidence of HTGAP in Australia in a single tertiary center, with a primary aim of routinely measuring triglyceride levels in order to establish incidence of HTGAP. A secondary aim was to describe a case series of patients with HTGAP with a focus on their co-morbidities, severity, and in-patient management.

## 2. Materials and Methods

A prospective cohort was collated over 18 months (July 2017 to January 2019) in John Hunter Hospital, a tertiary referral hospital in New South Wales, Australia. This study received ethics approval from the Hunter New England Human Research Ethics Committee for low and negligible risk (Reference number 17/06/21/5.10). Patients presenting to this center with acute pancreatitis (defined as 2 out of 3 of the following features: abdominal pain, serum lipase three-times the upper reference limit, radiological findings consistent with AP [19]) had their triglyceride levels measured as part of the study protocol. The acute surgical team managed all patients and referred to the endocrine team as appropriate. The study protocol included an automated reminder from the Clinical Chemistry department to the treating team sent at admission of any patient found to have a lipase greater than three-times the upper reference limit, prompting the admitting team to send a sample for measurement of triglyceride levels. HTGAP was defined as the co-existence of AP with triglycerides ≥11.2 mmol/L (1000 mg/dL) [1]. This cut-off is consistent with the Endocrine Society guidelines for severe hypertriglyceridemia and is considered to be a level that plausibly has a causative role in AP [1,20]. The patients’ demographic data, etiology of pancreatitis, intensive care unit (ICU) admission and in-patient management strategies were recorded prospectively using a standardized audit survey tool. Severe pancreatitis was defined pragmatically as the requirement for ICU admission or death. Admission to ICU was assessed individually by the ICU physician and acute surgical team. This is based on a hospital protocol that requires ICU review for AP patients on admission if they fulfil criteria for Systemic Inflammatory Response Syndrome or when a patient fulfils medical emergency criteria in the ward. Acute Physiology and Chronic Health Evaluation II (APACHE II) scores were calculated for those admitted to the ICU [21]. Patients diagnosed with post- Endoscopic retrograde cholangiopancreatography pancreatitis; malignancy-related obstruction or aged less than 18 years old were excluded. At the termination of the study, all International Classification of Diseases-10 hospital discharge codes (K850, K851, K852, K853, K858 and K859) within the study period were searched to identify missed patients, and the surgical audit database was interrogated to ensure all cases of AP had been included in the cohort.

Statistical analyses were performed using STATA version 14. Demographic and clinical data were reported as a number with percentages for categorical variables, mean ± standard error for continuous variables that show a normal distribution and median with interquartile range (IQR) for continuous variables without a normal distribution. The comparisons for categorical data were performed using Fisher’s exact test, and continuous data comparisons were performed using the Student’s *t*-test (parametric) and the Mann–Whitney test (non-parametric). Logistic regression was performed to estimate the association between HTGAP and ICU admissions. All two-sided *p*-values < 0.05 were considered significant.

## 3. Results

### 3.1. Comparison of the HTGAP and Non-HTGAP Cohorts

Of the 292 episodes of AP, 248 (85%) had triglycerides measured and were included in the cohort (Figure 1). HTGAP was diagnosed in 10 of 248 (4%) AP cases. A comparison of the demographic and clinical data in HTGAP and non-HTGAP cases is described in Table 1. Patients included in the HTGAP group were more likely to have a prior diagnosis of dyslipidemia (50% vs. 21%) or type 2 diabetes mellitus (60% vs. 15%). Severe hypertriglyceridemia was present at admission in the HTGAP group (median 51 mmol/L (IQR 46.8) vs. 1.3 mmol/L (IQR 0.9) Non-HTGAP).

### 3.2. HTGAP Case Series

Table 2 summarizes the in-patient admission for the HTGAP cases (*n* = 10), highlighting their clinical features, contributing factors and management. Within the HTGAP group, all cases had at least one secondary risk factor (type 2 diabetes mellitus, high-risk drinking or obesity) (Table 2). Type 2 diabetes (*n* = 6) and elevated body mass index (BMI > 25 kg/m^2^, *n* = 6) were common. Three HTGAP patients were already on lipid-lowering therapy prior to their acute admission, and two patients had previous admissions for HTGAP before July 2017. Alcohol misuse (*n* = 5) and gallstones (*n* = 3) may have directly contributed to the development of AP in the HTGAP group.

The HTGAP group had a trend towards longer hospital admissions compared to the non-HTGAP (median 6.7 vs. 2.8 days). ICU admissions were significantly increased (unadjusted odds ratio 16; 95% CI 4–62, *p* < 0.001) in the HTGAP group (5/10 vs. 14/238, *p* < 0.001) and constituted 26% (5/19) of total ICU admissions for AP. APACHE II scores for those admitted to ICU were similar in HTGAP compared to non-HTGAP patients (median (IQR): 10 (5) vs. 16 (12), *p* = 0.266) (Table 2). There was no significant difference in Charlson Comorbidity Index scores amongst HTGAP patients admitted to the ICU and not admitted to the ICU (median (IQR): 2 (0) vs. 1 (1), *p* = 0.382) (Table 2). There was one death (Case 2) due to multi-organ failure.

The endocrine team was asked to provide a recommendation for the management of hypertriglyceridemia in seven of the patients. Four patients (three with diabetes and one without diabetes) received intravenous insulin infusion therapy, which, together with fasting, resulted in rapid improvement of triglyceride levels by 65–77% within 24 h of admission (Table 2). These three patients with diabetes (Cases 1, 6 and 10) were all admitted to the ICU and required increments in insulin rate due to resistant hypertriglyceridemia and persistent hyperglycemia, respectively. They were then transitioned to subcutaneous insulin and a low-fat diet with resolution of hypertriglyceridemia over the following 3–5 days (Figure 2). One patient received low-dose subcutaneous insulin in view of her co-morbidities and stable clinical status, leading to a more gradual triglyceride improvement of 21% after 24 h (Table 2). Eight patients received dietitian input and were managed with a low-fat (<30 g/day) diet upon recommencement of oral intake. Eight patients received subcutaneous unfractionated heparin or low-molecular-weight heparin (enoxaparin) for deep venous thrombosis prophylaxis. Oral lipid-lowering therapy was commenced prior to discharge in six patients (statin *n* = 6, fibrate *n* = 5, omega-3 *n* = 1).

The only adverse event related to therapy was observed in Case 6, who had mild hypoglycemia (3.8 mmol/L) during intravenous insulin therapy (Figure 2) and was managed by increasing the intravenous glucose infusion. This case had very severe HTG of 115.5 mmol/L (or 10,221 mg/dL) on presentation, peaking at 163.3 mmol/L (or 14,451 mg/dL) seven hours after, leading to an increase in the insulin infusion rate from 4 units/h (0.05 units/kg/h) to 8 units/h (0.1 units/kg/h). This coincided with deteriorating hypoxia and tachycardia, leading to an ICU admission. Intervening with an insulin infusion and fasting resulted in a rapid reduction in triglyceride to 80.5 mmol/L (or 7123 mg/dL) five hours later (51% decline from peak triglyceride level), and further to 37.0 mmol/L (or 3274 mg/dL) at 24 h post-admission (77% decline from peak triglyceride level). Resuscitative intravenous fluids were administered as prescribed by the surgical team, and intravenous dextrose was titrated separately once blood glucose levels were <10 mmol/L, aiming for blood glucose levels between 5 and 10 mmol/L. An elective laparoscopic cholecystectomy was performed 2 months later in view of suspicious ultrasound findings suggesting concurrent gallstone etiology.

Case 9 was notable as they did not have a history of diabetes mellitus (HbA1c 5.6% or 38 mmol/mol) and was treated in a ward setting using fixed-rate intravenous insulin of 2 units/h (0.02 units/kg/h) administered concurrently with intravenous 4% dextrose + n/5 saline at 80 mL/h (Figure 2). The patient was closely monitored with hourly capillary glucose levels, aiming for blood glucose levels between 5 and 10 mmol/L and 4-hourly venous blood gases to ensure normokalaemia. Their triglyceride levels decreased by 70% after 24 h, without evidence of hypoglycemia.

## 4. Discussion

### 4.1. Incidence

We report hypertriglyceridemia-associated acute pancreatitis (HTGAP) in 4% of acute pancreatitis cases in our regional tertiary referral center, with an increased rate of ICU admission in the group with severe hypertriglyceridemia. The link between hypertriglyceridemia (HTG) and acute pancreatitis was first observed by Speck in 1865 [23]. Despite this long-standing observation, there is a paucity of high-quality data for the incidence and management of HTGAP. A systematic review by Adiamah et al. included 38 reports from 2006 to 2017, containing a minimum of ten HTGAP cases per study; the majority of included studies had an observational or case/control design [24], apart from a single-site, randomized controlled trial [25]. The reported incidence of HTGAP varied widely in these studies, between 2.3% and 53%, with the heterogeneity attributed to the variation in study design, the small numbers of participants and the lack of a standardized definition for HTGAP [24]. Specialized lipid clinics consistently report higher prevalence rates of HTGAP—for example, France 20% (TG defined as ≥11.2 mmol/L), Germany 19% (TG defined as ≥11.2 mmol/L) and Canada 15% (TG defined as ≥20 mmol/L) compared to 0.7–10% in population- or center-based studies [1,2,3,4,5,6,7,8,9]. The incidence in this prospective study is similar to that reported by Vipperla et al., who performed a large retrospective cohort study using uniform criteria to follow the natural history of 121 patients with HTGAP over 64 months [2]. In their study, 92% of cases with AP had TG measured; the incidence of HTGAP (TG defined as ≥5.6 mmol/L) was 6%, and almost half of this group had a peak TG ≥ 11.2 mmol/L [2].

### 4.2. Clinical Factors

Secondary causes of hypertriglyceridemia have been identified as common in the literature and were universally identified in the HTGAP individuals in this study. Secondary causes were similarly present in up to 46% of men and 94% of women with severe HTG (defined as >10 mmol/L) in the Copenhagen General Population Study (CPGS) and in 74% of cases in a Japanese study (defined as ≥11.2 mmol/) [26,27]. Type 2 diabetes, obesity and alcohol intake have been frequently associated with AP in both community-dwelling individuals and hospital data [26,27,28,29]. Poorly controlled type 2 diabetes mellitus has been found as the predominant clinical risk factor, irrespective of geographical location, e.g., 53% in Denmark, 74% in the United States and 30% in Japan [26,27,29]. This strong link is unsurprising, as in metabolic syndrome, the liver production of very low density lipoprotein (VLDL) is increased [18] and subsequent hydrolysis of VLDL by lipoprotein lipase increases circulating FFA [30]. This increase in FFA in combination with insulin resistance and decreased insulin signaling increases chylomicron secretion [30]. Hyperglycemia itself is known to stimulate chylomicron secretion, thereby potentiating the risk of HTG [18,30,31]. In fact, chylomicrons may be the key to causing AP, as seen in Familial Chylomicronemia syndrome (FCS) (defined as TG >10 mmol/L and absence of secondary HTG causes), which has higher rates of pancreatitis and recurrent pancreatitis compared to Multifactorial Chylomicronemia (MCM) (60% FCS vs. 6% MCM and 48% FCS vs. 3% MCM, respectively) [32].

It is notable that seven of the reported HTGAP cases in this study had either a concomitant gallstone or high-risk alcohol intake (Table 2). Dominguez-Munoz et al. performed serum triglyceride tests on 49 randomly selected patients with AP and identified other causes co-existing with HTG (defined as >20 mmol/l or 1772 mg/dL) in 47% of their cohort [33]. This was also observed in a prospective cohort study in Sweden which reported 31% of all AP cases with mild–moderate HTG (defined as 1.64–6 mmol/L) [3]. This raises the question of whether HTG plays a causal role in AP or is a biomarker of severity. Alcohol is known to be an independent cause of pancreatitis, yet it can also exacerbate HTG by competing with FFA for oxidation in the liver, thereby further increasing circulating FFA [34]. The notion that mild–moderate HTG (1.7–11.1 mmol/L or 150–999 mg/dL) is simply an epiphenomenon has been challenged by the finding of a statistically significant dose–response relationship that has been consistently reported with risk of AP at adjusted hazards ratio of 1.2 per 1 mmol/L (89 mg/dL) increments in Copenhagen (comparator ≥ 1 mmol/L, 89 mg/dL) and Sweden (≥0.86 mmol/L, 76 mg/dL) and 1.04 per 1.1 mmol/L (100 mg/dL) in Scotland (>1.7 mmol/L, 150 mg/dL) [3,35,36,37]. This was also observed in pre-clinical models of AP in which a dose-dependent increase in pancreatic amylase and lipase was observed after a 6-h triglyceride infusion in rat pancreases [13]. Yet, a specialized lipid clinic in Canada has found HTGAP to be causal only at a higher level of triglycerides >20 mmol/L (or 1770 mg/dL) [8].

### 4.3. Severity

Severe hypertriglyceridemia is associated with an increased severity of AP, a finding that has been demonstrated in both animal and human studies [2,7,38,39,40,41,42]. This is consistent with the finding of a significantly higher risk of ICU admissions in HTGAP. ICU admissions were used as a practical approach to measuring severity due to the observational nature of our study and clinician variation in using available scoring systems [43]. Of note, there was no significant difference in APACHE II or Charlson Comorbidity Index scores driving this higher risk of ICU admissions observed in our study. To date, there have been two meta-analyses performed to evaluate the relationship between triglyceride levels and severity of AP [38,39]. Concerns have been raised regarding the methodology of the first meta-analyses performed by Wang et al. (2017) as their analysis did not take into account the methodological differences in the definition and categorization of HTGAP [39]. A second meta-analysis published a year later evaluated studies based on hypertriglyceridemia definitions by the Endocrine Society [20,38]; this confirmed a significant increase in severity of acute pancreatitis with rising levels of triglycerides as compared to normal levels (OR 2.01 when TG ≥ 5.6 mmol/L, and OR 3.08 when TG ≥ 11.2 mmol/L) [38]. At TG ≥ 11.2 mmol/L, there was a significant increased risk of persistent organ failure and ICU admission, but without a significant increase in mortality compared to those with normal TG levels [38]. Other studies, however, have not demonstrated a correlation with absolute triglyceride levels and disease severity [7].

### 4.4. Acute Management of HTGAP

The initial management of HTGAP is common to other forms of AP [18,24]. It is recommended that triglyceride levels are measured in all presentations of acute pancreatitis for severity prognostication, considering that other causes, such as gallstones, may also be present [1,3]. Triglyceride levels should be monitored early in the course of AP as levels can decline rapidly within 48–72 h, leading to underestimation of its true prevalence [33,37]. Fluid resuscitation with intravenous (IV) glucose is not recommended as short-term infusions of IV glucose may induce hyperglycemia and increase intestinal apolipoprotein B48, a component of chylomicron, and VLDL secretion in humans [31].

The therapeutic options for acutely lowering triglyceride levels in HTGAP include IV insulin, heparin and lipid apheresis [18]. Insulin acts as a cofactor for lipoprotein lipase (LPL) and activates peripheral endothelial LPL to hydrolyze circulating triglycerides to monoglycerides and free fatty acids, whilst inhibiting adipocyte hormone-sensitive lipase to prevent free fatty acid production and, consequently, decreasing production of VLDL from the liver [44]. LPL is normally bound to heparan sulfate proteoglycans (HSP) on the vascular endothelium, where it hydrolyses TG [45]. IV unfractionated heparin, structurally similar to HSP, rapidly increases circulating LPL and forms the active heparin apoenzyme complex to promote a decrease in TG levels [46]. However, LPL activity has been observed to decrease exponentially due to destruction of this active complex by liver heparinase [46,47]. Concerns have been raised about using heparin in the management of HTGAP because of an increased bleeding risk, for example, in patients with pancreatic necrosis, and the observed temporary depletion of LPL, if the hepatic degradation of LPL exceeds its production and release, resulting in rebound HTG [48,49,50]. It is notable that evidence supporting the use of insulin and heparin is mostly limited to small case-control studies and case series [24]. There is one single-center, randomized controlled trial (*n* = 66 HTGAP) that compared high-volume hemofiltration (HVHF) against combination low-molecular-weight heparin (LMWH 4000 IU every 12 h for 3 days) and insulin (titrated via a pump to maintain blood glucose level between 7.8 and 11.1 mmol/L (140–200 mg/dL)) [25]. HVHF achieved target TG (defined as <5.6 mmol/L) in 9 h compared to 48 h for the combination LMWH and insulin group [25]. However, despite the observed biochemical advantage of HVHF, there was a doubling in the cost of treatment and, importantly, a significantly higher incidence of persistent organ failure [25].

IV insulin therapy, when combined with fasting, is an effective management option as it has been shown to reduce triglyceride levels by 87% in 24 h, as compared to a 40% TG reduction with IV insulin alone, and 23% with subcutaneous insulin alone [51]. Four patients within this case series received IV insulin for management of HTGAP; their triglyceride levels fell to the target (defined as <11.2 mmol/L) within 3–5 days (Figure 2), consistent with the reported timeframes in the literature [9,52]. It is important to recognize that as yet, a significant outcome benefit has not been demonstrated after intervention with IV insulin therapy compared to standard conservative management; however, it is a safe treatment modality if close monitoring for hypoglycemia is achieved [9]. IV insulin therapy can be utilized for management of HTGAP in those without diabetes, provided that intravenous dextrose is used concurrently to maintain normoglycemia [53], as observed in Case 9 (Figure 2). Further research is needed to demonstrate clinical benefit in morbidity and mortality outcomes. As a result of our experiences in this study, a treatment algorithm for the management of hypertriglyceridemia in HTGAP for those with and without diabetes has been proposed for our hospital (Appendix A, Figure A1).

Lipid apheresis rapidly reduces TG by 50–80% [18]. However, despite its efficacy, lipid apheresis has not been shown to deliver clinical benefits, other than in one study that reported a reduction in hospital length-of-stay after treatment for those with extreme HTG (defined as >56.5 mmol/L) [1,24,54]. Questions remain regarding the appropriate timing and frequency that might be required for an appreciable clinical benefit with lipid apheresis [1,18,24]. Randomized controlled trials are lacking in this area and current studies have heterogenous methodology [1,24]. Rebound HTG can occur after initial treatment if the underlying cause of pancreatitis is not addressed.

The main strength of our study is the prospective design that allows a reliable estimation of the incidence of HTGAP. Our incidence rate of 4% is much higher than the rate of 0.8% derived from an earlier observational audit performed retrospectively in the same center prior to the introduction of this study protocol [55]. This can be attributed to the lack of triglyceride measurements as a usual standard of care in AP [55]. The current study also reflects real-life management of HTGAP, with the surgical team leading clinical management for AP.

The size of the HTGAP cohort and observational design of this study limit the ability to assess the efficacy of the management interventions for HTGAP. The limitations of an observational study led to some missing data points, particularly for HbA1c and body mass index. The HTGAP cohort contained few patients with HTG as a sole potential factor that could have contributed to the development of AP, and we were not able to perform logistic regression modelling to evaluate associated clinical risk factors for ICU admissions in HTGAP. This likely reflects ‘real life’, in which there are multiple triggers.

## 5. Conclusions

HTGAP occurred in 4% of AP in a single Australian center and was associated with higher risk of ICU admission. Hypertriglyceridemia is a marker of severity for AP and could, in itself, be a causative factor. In our study, fasting, low-fat diet and intravenous insulin therapy were used to reduce triglyceride levels. At the present time, the use of IV insulin in patients without diabetes remains experimental [24], but it can be safely administered with close monitoring in situations where lipid apheresis is not preferable. Larger randomized controlled trials investigating the efficacy of IV insulin (with or without fasting), unfractionated heparin and lipid apheresis compared to conservative management, alongside direct comparisons or combination therapy, are required to determine the interventions that provide the optimal clinical outcomes.

## Figures and Tables

**Figure 1 jcm-09-03954-f001:**
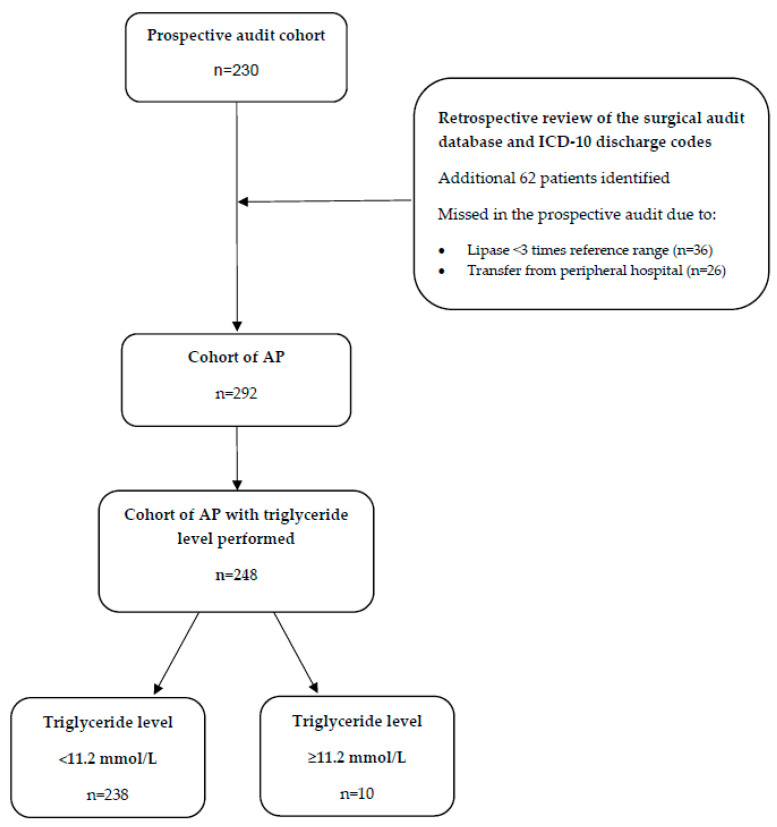
Participant flow chart. AP: acute pancreatitis.

**Figure 2 jcm-09-03954-f002:**
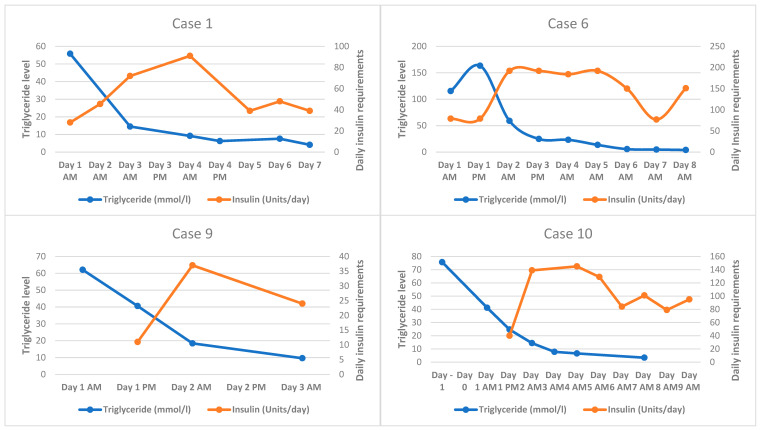
Triglyceride trends with insulin therapy. **Case 1:** Intravenous insulin infusion of 4 units/h (0.05 units/kg/h), then transitioned to subcutaneous insulin on Day 4. **Case 6**: Intravenous insulin infusion of 8 units/h (0.1 units/kg/h) and required 10% dextrose infusion due to hypoglycemia. Transitioned to subcutaneous insulin on Day 7. **Case 9**: Intravenous insulin infusion of 2 units/h (0.02 units/kg/h). Insulin ceased completely on Day 3 as no known diabetes (HbA1c 38 mmol/mol, 5.6%). **Case 10**: Intravenous insulin infusion of 8 units/h (0.1 units/kg/h), then transitioned to subcutaneous insulin on Day 7.

**Table 1 jcm-09-03954-t001:** Demographic and clinical data comparing hypertriglyceridemia-associated acute pancreatitis (HTGAP) and non-HTGAP groups.

Parameter	Overall	Non-HTGAP	HTGAP	*p*-Value
n	248	238	10	
Age				
Years ± SEM	53.9 ± 1.2	54.2 ± 1.2	48.7 ± 2.4	0.37
Male				
*n* (%)	117 (47.2)	111 (46.6)	6 (60)	0.50
Body mass index				
*n*/total (%) ^+^	80/248 (32)	74/238 (31)	6/10 (60)	0.86
mean ± SEM, kg/m^2^	31.0 ± 0.8	31.1 ± 0.9	30.6 ± 1.6	
Smoker *				
*n*/total (%)	68/190 (36)	64/180 (34)	4/10 (40)	0.75
Alcohol risk				
High **	57(33)	52 (32)	5 (50)	0.42
Low	41 (25)	40 (25)	1 (10)	
Non-drinker	72 (42)	68 (43)	4 (40)	
Known dyslipidemia ^‡^				
*n* (%)	54 (22)	49 (21)	5 (50)	0.04
Diabetes ^‡^				
Type 1	5 (2.0)	5 (2.1)	0 (0.0)	0.01
Type 2	42 (17.1)	36 (15.3)	6 (60.0)	
Gestational	1 (0.4)	1 (0.4)	0 (0.0)	
Nil	198 (80.5)	194 (82.2)	4 (40.0)	
Lipase level, IU/L	1154	1145	1731	0.67
Median (IQR)	(2915)	(3065)	(1725)	
Triglyceride level, mmol/L	1.3	1.3	51	<0.001
Median (IQR)	(1.1)	(0.9)	(46.8)	
Length of stay, days				
Median (IQR)	2.9 (3.7)	2.8 (3.7)	6.7 (9.8)	0.46
ICU admission				
*n* (%)	19 (8)	14 (6)	5 (50)	<0.001
Mortality				
*n* (%)	7 (3)	6 (3)	1 (10)	0.25

^+^ Number and percentage of cohort with available body mass index data. * Smoker defined as current or ceased for <12 months. ** High alcohol risk defined as >10 standard drinks/week and/or binge-drinker defined as >4 standard drinks/day [22]. ^‡^ Clinician diagnosis and/or medication.

**Table 2 jcm-09-03954-t002:** HTGAP case series—clinical risk factors, concurrent etiology and management. Cases presented in chronological order of presentation during the study.

	Gender	Age (Years)	Triglyceride (TG) Level on Admission (mmol/L)	Pre-Admission Lipid Treatment	Previous AP	BMI kg/m^2^	T2DM (HbA1c mmol/mol, %)	Smoker	Concurrent Etiology		ICU Admission (APACHE II, Charlson Comorbidity Index)	Endocrine Input	In-Patient Management of HTGAP	TG Level after 24 h, mmol/L (% Decline from Admission)
									Gallstones on Imaging	Alcohol **				
1	Female	53	55.8	Statin Fenofibrate	Y	32	Y (45, 6.3)	Y	Y	No risk	Y (16, 2)	Day 3 of admission	Inotropic support NBM, then <30 g fat daily Intravenous insulin infusion increased from 3 to 4 units/h (0.05 units/kg/h) after 48 h Statin, fenofibrate Laparoscopic cholecystectomy Subcutaneous enoxaparin DVT prophylaxis	NA †
2 ∞	Male	55	75.6	Statin	Y	-NA	N	N	Previous resection	High risk	Y (14, 3)	N	Inotropic support Nasogastric feeding Supportive therapy for multi-organ failure Subcutaneous heparin DVT prophylaxis	NA
3	Female	52	15.3	N	N	30	N	N	N	High risk	Y (8, 2)	N	NBM, then <30 g fat diet Nil specific therapy for triglycerides Subcutaneous heparin DVT prophylaxis	NA
4	Female	59	11.3	Ceased statin 1 month ago	N	26	Y ‡ (46, 6.4)	Y	Previous resection	No risk	N (NA, 7)	Y	NBM, then <30 g fat diet Subcutaneous basal-bolus insulin Statin Fenofibrates not commenced due to end-stage kidney disease (peritoneal dialysis). Omega-3 tablets ceased due to dyspepsia Subcutaneous heparin DVT prophylaxis	8.9 (21%)
5	Male	34	52.4	N	N	37	Y (NA)	N	N	High risk	N (NA, 1)	N	NBM, then <30 g fat diet Intravenous fluids Subcutaneous enoxaparin DVT prophylaxis	NA
6	Male	48	115.5 ^	Statin Gemfibrozil Ezetimibe	N	NA	Y (75, 9.0)	Y	Possible	No risk	Y (10, 2)	Y	NBM, then <30 g fat daily Intravenous fluids Intravenous insulin infusion increased from 4 to 8 units/h (0.1 units/kg/h) after 5 h Statin, fenofibrate, ezetimibe Subcutaneous enoxaparin DVT prophylaxis	37.0 (77%) #
7	Female	50	49.6	N	N	NA	N	Y	Possible	High risk	N (NA, 1)	Y	NBM, then <30 g fat daily Intravenous fluids and electrolyte replacement Statin not commenced due to liver derangement and spontaneous improvement of TG after 24 h	5.9 (88%)
8	Male	52	11.2	Self-ceased statin and fenofibrate 4 years ago	Y	NA	Y (52, 6.9)	N	N	Low risk	N (NA, 2)	Y	NBM, then <30 g fat daily Intravenous fluids Statin, fenofibrate, omega-3	3.6 (68%)
9	Male	38	62.1	N	N	27	N (38, 5.6)	N	N	High risk	N (NA, 0)	Y	NBM, then <30 g fat daily Intravenous fluids Intravenous insulin infusion 2 units/h (0.02 units/kg/h) Statin, fenofibrate Subcutaneous enoxaparin DVT prophylaxis	18.4 (70%)
10	Male	49	41.2	N	N	31	Y * (104, 11.7)	N	N	No risk	Y (9, 1)	Y	NBM, then <30 g fat daily Intravenous fluids Intravenous insulin infusion increased from 4 to 6 units/h (0.06 units/kg/h) after 24 h Statin, fenofibrate Subcutaneous enoxaparin DVT prophylaxis	14.4 (65%)

T2DM = type 2 diabetes mellitus; NBM = nil by mouth; DVT = deep venous thrombosis; Y = Yes; N = No; NA = Not available. ^†^ Triglyceride level not performed 24 h post admission. Triglyceride level 48 h post admission was 14.5 mmol/L (74% reduction from admission) whilst fasting with insulin infusion at 3 units/h (gradually uptitrated by ICU team). Insulin infusion increased to 4 units/h (0.05 units/kg/h) after endocrine review on day 3 with reduction in TG by a further 21% to 11.4 mmol/L. ^∞^ Death from multiorgan failure. ^‡^ Not on treatment. Insulin ceased few months ago due to hypoglycemia. ^ Peak triglyceride level 163.3 mmol/L 7 h post presentation. ^#^ Percentage decline calculated from peak triglyceride level (163.3 mmol/L). * New diagnosis of diabetes. ** High alcohol risk defined as >10 standard drinks/week and/or binge-drinker defined as >4 standard drinks/day [22].

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
