# Peer review of "Incidence and Management of Hypertriglyceridemia-Associated Acute Pancreatitis: A Prospective Case Series in a Single Australian Tertiary Centre"

_jcm, 2020, doi:10.3390/jcm9123954_

Round 1
Reviewer 1 Report
I have the oportunity to review the paper entitled "Incidence and management of hypertriglyceridaemia- associated acute pancreatitis: a prospective case series in a single Australian tertiary centre", written by Hong Lin Evelyn Tan and co-workers.
It shows clinical data of a serie of ten patients with acute hyperttryliceridemic pancreatitis and compare them with non-HTG AP. The most impressive results is that 50% of this subjects were admitted to ICU, which is remarkably high in comparisons with other etiologies of AP.
MAJOR COMMENT: I accept that a pragmatic way to define the severity of an AP is the need to be admitted in ICU. However, nothing is said about the criteria that a patient with AP should fullfil in the center were the study was carried out to enter in ICU. Furthermore, that definition impedes the comparison with other series, where severity is shown as Ransom's criteria or, even better, APACHE score. APACHE score is performed routinely in every ICU worldwide and should be added to the manuscript. I think the authors will have the oportunity to add this information in subject admitted to ICU, both ih HTGAP and nonHTGAP groups.
In line 86, it is mentioned that a logistic regression analyses was performed to estimate the association between HTGAP and ICU admission; however, I don't see this data in results or discussion section. Please add the APACHE score to the independent variables.
MINOR COMMENTS:
- Results, 3.2. case series: I think that the subjects with HTGAP are so well described in Table 2 that the text should be shortened.
- Discussion: may be shortened too. Regarding the role of HTG in the pathogenesis and etiology of AP it should be noted that are not triglycerides causing AP, are the Chylomicrons that are usually present with levels of TG over 1000 mg/dL. In fact, individuals with familial chylomicronemia syndrome (low BMI, Tg > 2000 mg/dL, low levels of VLDL and no insulin resistance or alcohol conpsumption) are at the highest risk for recurrent HTGAP.
- As the authors mention in the conclusion, we need RCT comparing IV insulin, unfractionated heparin, plasmapheresis vs usual care (NBM, analgesics, IV fluids) to choose the best therapeutical option in HTGAP. Despite of this statement, in the algorithm proposed they are going to consider IV insulin infusion in non diabetic patients. Why?.
Author Response
Response to Reviewer
Article Title: Incidence and management of hypertriglyceridaemia-associated acute pancreatitis: a prospective case series in a single Australian tertiary centre
Reviewer 1
Thank you for reviewing our manuscript. We have addressed your feedback below. Alterations to the text are in red.
- MAJOR COMMENT: I accept that a pragmatic way to define the severity of an AP is the need to be admitted in ICU. However, nothing is said about the criteria that a patient with AP should fullfil in the center were the study was carried out to enter in ICU.
Admission to ICU was assessed individually by the acute surgical team and ICU physician. The hospital has a protocol that requires ICU review on admission if they fulfil criteria for Systemic Inflammatory Response or when a patient fulfils medical emergency criteria in the ward. This is described in the methods Page 2 lines 76-79 “Admission to ICU was assessed individually by the ICU physician and acute surgical team. This is based on a hospital protocol that requires ICU review for AP patients on admission if they fulfil criteria for Systemic Inflammatory Response Syndrome or when a patient fulfils medical emergency criteria in the ward.”
- Furthermore, that definition impedes the comparison with other series, where severity is shown as Ransom's criteria or, even better, APACHE score. APACHE score is performed routinely in every ICU worldwide and should be added to the manuscript. I think the authors will have the opportunity to add this information in subject admitted to ICU, both in HTGAP and non HTGAP groups.
The authors note that the primary aim of this prospective study was to identify the incidence of HTGAP and therefore the study was not powered to compare the severity of pancreatitis between the two groups. We agree that the finding of an increased ICU admission is interesting and we have reported the Odds Ratio in results section page 3 lines 110-111 “ICU admissions were significantly increased (unadjusted odds ratio 16, 95% CI 4-62, p<0.001) in the HTGAP group (5/10 vs. 14/238, p<0.001) and constituted 26% (5/19) of total ICU admissions for AP.”
We agree with the reviewer that a measure of severity for those patients admitted to ICU may be of interest to the readers. The methods section has been adjusted in page 2 line 79-80 “Acute Physiology And Chronic Health Evaluation II (APACHE II) scores were calculated for those admitted to ICU [21].” Kruskal-Wallis test has been used to measure p-values as included in the methods section page 2 line 90 “The comparisons for categorical data were performed using Fisher’s exact test, and continuous data comparisons were performed using student t-test (parametric) and Kruskal-Wallis test (non-parametric).”
Individual APACHE II scores for HTGAP patients have been added to Table 2, ICU column (Page 6-7). A comparison of the APACHE II score for the patients with HTGAP and non-HTGAP is reported in the results page 3 lines 112-113 “APACHE II scores for those admitted to ICU were similar in HTGAP compared to non-HTGAP patients (median [IQR]: 10 [5] vs 16 [12], p=0.518) (Table 2).”
This is included in the discussion page 11 lines 236-238 “Of note, there was no significant difference in APACHE II or Charleson Comorbidity Index scores driving this higher risk of ICU admissions observed in our study.”
- In line 86, it is mentioned that a logistic regression analyses was performed to estimate the association between HTGAP and ICU admission; however, I don't see this data in results or discussion section. Please add the APACHE score to the independent variables.
Odds ratio for ICU admissions was described in results section page 3 lines 110-111 (as above) and abstract section page 1 line 24.
Logistic regression modelling was not performed in view of small numbers of participants with HTGAP. We have highlighted this in the limitations of our study in discussion page 12 lines 310-311 “The HTGAP cohort contained few patients with HTG as a sole potential factor that could have contributed to the development of AP and we were not able to perform logistic regression modelling to evaluate associated clinical risk factors for ICU admissions in HTGAP. This likely reflects ‘real-life’, in which there are multiple triggers.”
- Results, 3.2. case series: I think that the subjects with HTGAP are so well described in Table 2 that the text should be shortened.
Two cases of HTGAP were specifically mentioned in the results section to highlight our management of HTGAP in (1) a diabetes patient who had a complication from treatment of HTGAP, and (2) a scenario where clinicians are less comfortable with of a patient without diabetes who received insulin infusion to manage HTGAP.
Importantly, we wished to emphasise that resuscitative fluids are guided by the surgical team. HTGAP management, albeit very important, should be managed alongside usual routine of care of the acute pancreatitis patient. We also wanted to emphasise the importance of close monitoring of patients whilst on insulin infusion to prevent complications of hypoglycaemia.
However, we agree that the description could have greater brevity to maintain clarity for the reader: (Page 3-4, lines 131-156)
“The only adverse event related to therapy was observed in Case 6, who had mild hypoglycaemia (3.8 mmol/L) during intravenous insulin therapy (Figure 2) and was managed by increasing the intravenous glucose infusion. This case had very severe HTG of 115.5mmol/L (or 10,221mg/dL) on presentation, peaking at 163.3 mmol/L (or 14,451mg/dl) seven hours after, leading to an increase in the insulin infusion rate from 4 units/h (0.05 units/kg/h) to 8 units/h (0.1 units/kg/h). This coincided with deteriorating hypoxia and tachycardia leading to an ICU admission. Intervening with an insulin infusion and fasting resulted in a rapid reduction of triglyceride to 80.5mmol/L (or 7123mg/dL) five hours later (51% decline from peak triglyceride level), and further to 37.0 mmol/L (or 3274mg/dL) at 24 hours post admission (77% decline from peak triglyceride level). Resuscitative intravenous fluids were administered as prescribed by the surgical team, and intravenous dextrose was titrated separately once blood glucose levels were <10mmol/L aiming for blood glucose levels between 5-10 mmol/L. An elective laparoscopic cholecystectomy was performed 2 months later in view of suspicious ultrasound findings suggesting concurrent gallstone aetiology.
Case 9 was notable as he did not have a history of diabetes mellitus (HbA1c 5.6% or 38 mmol/mol) and was treated in ward-setting using a fixed rate intravenous insulin of 2 units/hour (0.02 units/kg/h) administered concurrently with intravenous 4% dextrose + n/5 saline at 80ml/h (Figure 2). The patient was closely monitored with hourly capillary glucose levels aiming for blood glucose levels between 5-10 mmol/L and 4-hourly venous blood gases to ensure normokalaemia. Their triglyceride levels decreased by 70% after 24 hours without evidence of hypoglycaemia.
- Discussion: may be shortened too.
Thank you. We have read through the discussion and believe that this section provides a comprehensive review of current literature and places the results of our study into context. The heterogeneity of study designs and lack of HTGAP definition meant that different triglyceride thresholds and comparators are used which we believe is important to highlight to the audience. To facilitate readers, we have divided the discussion into sub-sections of “Incidence”, “Clinical factors” “Severity”, and “Acute Management of HTGAP”.
- Regarding the role of HTG in the pathogenesis and etiology of AP it should be noted that are not triglycerides causing AP, are the Chylomicrons that are usually present with levels of TG over 1000 mg/dL. In fact, individuals with familial chylomicronemia syndrome (low BMI, Tg> 2000 mg/dL, low levels of VLDL and no insulin resistance or alcohol consumption) are at the highest risk for recurrent HTGAP.
Thank you for your feedback. We have added in the discussion section at page 10 lines 208-212 “In fact, chylomicrons may be the key to causing AP as seen in Familial Chylomicronemia syndrome (FCS) (defined as TG>10mmol/L and absence of secondary HTG causes), which has higher rates of pancreatitis and recurrent pancreatitis compared to Multifactorial Chylomicronemia (MCM) (60% FCS vs 6% MCM and 48% FCS vs 3% MCM respectively) [32].”
- As the authors mention in the conclusion, we need RCT comparing IV insulin, unfractionated heparin, plasmapheresis vs usual care (NBM, analgesics, IV fluids) to choose the besttherapeutical option in HTGAP. Despite of this statement, in the algorithm proposed they are going to consider IV insulin infusion in non diabetic patients. Why?.
The statement has been amended to read page 12 lines 317-322 “At the present time, the use of IV insulin in patients without diabetes remains experimental [24] but can be safely administered with close monitoring in situations where lipid apheresis is not preferable. Larger randomized controlled trials investigating the efficacy of IV insulin (with or without fasting), unfractionated heparin and lipid apheresis compared to conservative management, alongside direct comparisons or combination therapy, are required to determine the interventions that provide the optimal clinical outcomes.”
Thank you.
Reviewer 2 Report
Dear Authors,
I have read with interest the article from Dr Tan et al describing a prospective case series of hypertriglyceridemia-associated acute pancreatitis (HTGAP).
The article is of interest being this disease often overlooked in the context of acute pancreatitis. In addition, to date data related to HTGAP epidemiology and management are still scant and therefore this work might shed some light on this topic.
In particular, the authors designed a prospective study, dosing tryglicerids in all the patients referred to their tertiary care centre for acute pancreatitis and evaluating outcomes.
There few things that need to be addressed:
- while ICU admittance can be use as a proxy for disease severity, patients' coexisting condition should be underlined. indeed, ICU admission might well be riven by comorbidities rather than AP severity.
- page 3 line 118. 'subcutaneous unfractionated heparin'. Should that be EV?
- few sentences should be rewieved in terms of syntaxis
Author Response
Response to Reviewer
Article Title: Incidence and management of hypertriglyceridaemia-associated acute pancreatitis: a prospective case series in a single Australian tertiary centre
Reviewer 2
Thank you for reviewing our manuscript. We have addressed your feedback below. Alterations to the text are in red.
- while ICU admittance can be use as a proxy for disease severity, patients' coexisting condition should be underlined. indeed, ICU admission might well be riven by comorbidities rather than AP severity.
We acknowledge that patients co-existing morbidities would change the likelihood of requiring ICU admission. We therefore reviewed the Charlson Comorbidity Index (CMI) for our case series. Interestingly, the Index was not significantly higher in the five patients with HTGAP admitted to ICU (median 2, IQR 0) compared to the five patients who did not require ICU admission (median 1, IQR 1 p=0.420). The CMI score has been reported in Table 2, ICU column (page 6-7). A comparison of the CMI score for the HTGAP patients requiring or not requiring ICU care is reported in the results page 3 lines 113-115 “There was no significant difference in Charleson Comorbidity Index amongst HTGAP patients admitted to ICU and not admitted to ICU (median [IQR]: 2 [0] vs 1[1], p=0.420) (Table 2).”
Kruskal-Wallis test has been used to measure p-values as included in the methods section page 2 line 90 “The comparisons for categorical data were performed using Fisher’s exact test, and continuous data comparisons were performed using student t-test (parametric) and Kruskal-Wallis test (non-parametric).”
This is also included in the discussion page 11 lines 236-238 “Of note, there was no significant difference in APACHE II or Charleson Comorbidity Index scores driving this higher risk of ICU admissions observed in our study.”
- page 3 line 118. 'subcutaneous unfractionated heparin'. Should that be EV?
Thank you for clarifying. Did you mean intravenous? I can confirm that subcutaneous, and not intravenous, unfractionated heparin was used for deep venous thrombosis prophylaxis.
- few sentences should be reviewed in terms of syntaxis
Thank you, we have proof read the manuscript and consider it reads well.
Round 2
Reviewer 1 Report
I have reviewed the second version of the manuscript. The authors corrected my major comment on the severity of AP using the APACHE score and the protocol driving patients to ICU.
Now, I really don't understad why the authors include Charleson (Charlson?) comorbidities index, which is an score to predict life expectancy.
In M&M they used Student t-test or Kruskal-Wallis (shouldn't it be Mann-Whitney?).
